# Potassium Intake and Bone Health: A Narrative Review

**DOI:** 10.3390/nu16173016

**Published:** 2024-09-06

**Authors:** Veronica Abate, Anita Vergatti, Nadia Altavilla, Francesca Garofano, Antonio Stefano Salcuni, Domenico Rendina, Gianpaolo De Filippo, Fabio Vescini, Lanfranco D’Elia

**Affiliations:** 1Department of Clinical Medicine and Surgery, University Federico II, 80131 Naples, Italy; veronica.abate@unina.it (V.A.); anita.vergatti@unina.it (A.V.); nadia.altavilla@unina.it (N.A.); fr.garofano@studenti.unina.it (F.G.); lanfranco.delia@unina.it (L.D.); 2Unit of Endocrinology and Metabolism, University-Hospital S. M. Misericordia, 33100 Udine, Italy; salcuni.antonio@libero.it (A.S.S.); fabio.vescini@uniud.it (F.V.); 3Assistance Publique-Hôpitaux de Paris, Hôpital Robert Debré, Service d’Endocrinologie et Diabétologie, 75019 Paris, France; gianpaolo.defilippo@aphp.fr

**Keywords:** potassium, potassium salt, metabolic acidosis, bone mineral density, western diet

## Abstract

Potassium is a cation involved in the resting phase of membrane potential. Diets rich in fresh fruit and vegetables, whole grains, dairy products, and coffee have high potassium content. The shift from a pre-agriculture diet to today’s consumption has led to reduced potassium intake. Indeed, the Western diet pattern is characterized by a high daily intake of saturated fats, sugars, sodium, proteins from red meat, and refined carbohydrates with a low potassium intake. These reductions are also mirrored by high sodium intakes and a high consumption of acid-generating food, which promote a chronic state of low-grade metabolic acidosis. The low-grade metabolic acidosis is a cause of the bone-wasting effect. Therefore, a long-standing acidotic state brings into play the bone that contributes to the buffering process through an increase in osteoclastic resorption. In consideration of this background, we carried out a review that focused on the pathophysiological mechanisms of the relationship between dietary potassium intake and bone health, underlining the detrimental effects of the Western dietary patterns characterized by low potassium consumption.

## 1. Introduction

Potassium (K) is a highly reactive alkali metal which makes up 2.4% of the mass of the Earth’s crust. It is a vital positive ion, with 19 as its atomic number and 39.1 as its atomic mass. A 70 kg adult male is estimated to have a total K content of around 135 g [1]. After ingestion of food and water, K is absorbed by the small intestine with a passive mechanism [2]. On the cellular surface of the distal colon, K is excreted in exchange for sodium, and it may also be reabsorbed in the colon through active transport, which can be of importance during K deprivation [2]. The remaining K is eliminated with stools.

During a meal, the insulin-mediated cellular intake of K is increased, thus representing the K storage of the body, while in starving conditions, the cells release K to maintain a stable serum concentration of this ion [3]. Only two percent of the total K is extracellularly located, at a concentration ranging from 3.5 to 5 mmol/L. Indeed, a blood concentration of K above or below the normal amount can cause muscle weakness and cardiac arrhythmias. K takes part in the resting phase of membrane potential. Deviances of serum K levels in both directions are associated with plasma membrane hyper- or de-polarizations which may lead to a disturbed excitation of muscle and nerve cells, with the myocardium being especially vulnerable [4]. For this reason, the concentration of K is carefully regulated to maintain serum concentration stability, mostly by the kidneys through urinary excretion in healthy condition [5]. Because of tight homeostatic mechanisms, blood K concentrations and total body K content are only minimally affected by variations in dietary K intake.

Thirty-fold K concentrations are intracellularly located by the influence of the acid–base status, hormones, such as insulin and catecholamines, osmolality, and/or osmotic pressure. Muscle cells alone contain 70% of bodily K, followed by bone, liver, skin, and red blood cells [6].

K is filtered by the glomerulus at a rate of 756 mmol/day, considering an estimated glomerular filtration rate of 180 L/day and a K serum concentration of 4.2 mmol/L [7] (Figure 1).

During the ingestion of food and water, K is absorbed by the small intestines through the passive mechanism of absorption [8]. On the contrary, during deprivation, it is reabsorbed by the colon through the action of active transport mechanism. On the cellular surface of the distal colon, one K is excreted in exchange for one sodium (Na). In the kidneys, K is reabsorbed along all the nephron segments in the kidneys through active and passive mechanisms. (i) Proximal tubule: most of the filtered K is reabsorbed primarily by passive mechanisms in the proximal tubule. It takes place in a paracellular pathway (dotted line). (ii) Loop of Henle: it happens through paracellular and transcellular pathways. A basolateral sodium/potassium-ATPase pump (Na/K-ATPase) generates a positive potential, which provides more K for a sodium–potassium–chlorine cotransport (NKCC 2). This last co-transport allows for the reabsorption of one K with one Na and two chlorine (Cl) ions. Subsequently, the apical renal outer medullary K channel (ROMK) offers K for the NKCC2 transport mediation. (iii) Distal tubule: the amiloride-sensitive epithelial sodium channel (ENaC) and ROMK are overall distributed in the distal renal tubule. ENaC regulates the final excretion of K via the ROMK. The last regulators are represented by hydrogen/potassium-ATPase (H/K-ATPase) and Na/K-ATPase via aldosterone. Aldosterone increases basolateral Na/K-ATPase expression and activity. Lastly, K is also reabsorbed by H/K-ATPase.

For this reason, the daily K intake can be assessed by the measurement of K levels in 24 h urine collection. A factor of 1.30 has been chosen for converting dietary intake of K to K levels in 24 h urine collection (around 70% of K ingested is excreted in the urine) [9,10], and it is considered currently reliable [11,12].

## 2. Potassium Sources

In most countries worldwide, the habitual average of dietary K intake is largely below the WHO recommended minimum target per day (90 mmol per day) [13], although the international guidelines for the prevention and management of cardiovascular risk recognize the cardiovascular benefits associated with a high K intake [14,15].

K can be found in starchy roots or tubers, vegetables, fruits, whole grains, dairy products, and even in coffee. Fresh foods such as potatoes, vegetables, fruit, and legumes have the highest concentrations of K (Table 1 shows the K content in foods per 100 g of each type of food).

Given the widespread availability of K in so many different types of food, hypokalemia resulting from insufficient dietary intake is rare and may be associated with severe hypocaloric diets.

K in food is combined with a weak acid as K-salt. The most popular K-salts are K-L-ascorbate, magnesium-K-citrate, K-iodide, K-iodate, K-bicarbonate, K-carbonate, K-chloride, K-citrate, K-gluconate, K-glycerophosphate, K-lactate, K-hydroxide, K-salts of orthophosphoric acid, and K-fluoride [15]. Among them, K-chloride is used as a preservative during food processing. Many K-salts have long been used as supplements, in order to reduce the risk of complications of many medical conditions, such as nephrolithiasis [22]. K-citrate, K-bicarbonate, and K-chloride are the most used salts [16].

A sufficient amount of K may be provided by diets that include a high intake of vegetables and fruits, such as the Mediterranean (Med) diet [23], the Dietary Approaches to Stop Hypertension (DASH) diet [23], and the vegetarian diet [24].

Although there is no current (unambiguous) definition of the Med diet, it is recognized as the “traditional diet” adopted by populations residing in areas bordering the Mediterranean Sea. It is mostly characterized by high consumption of vegetables, legumes, and fruits (also known as plant-based food), providing a high amounts of K and a low amounts of sodium. The Med diet also requires high consumption of cereals, fish and dairy products, the use of olive oil for cooking as a source of fat, and low consumption of refined carbohydrates and animal proteins [25].

The DASH diet has been promoted in the United States to prevent and treat high blood pressure induced by the high sodium and low K intake from the WD [26]. The DASH diet resembles many of the characteristics of the Med diet. Indeed, it is composed of high content of fruits and vegetables, dairy foods, wholegrain cereals, nuts and seeds intakes, but restricted in saturated fats and refined sugars. In this way, compared to the WD, the DASH diet supplies higher K, calcium, magnesium, fiber, and proteins, as well as lower saturated fats [26,27,28].

The vegetarian diet is characterized by completely replacing animal proteins in the conventional diet [28]. It contains greater amounts of K, phosphorus, and calcium, but lower amounts of long-chain *n*-3 polyunsaturated fatty acids and vitamin B12 than the conventional diet. It is very similar to the plant-based diet, in which the consumption of small quantities of animal-based foods, including milk, eggs, meat, and fish, is allowed. However, both diets are characterized by high intake of fruit, vegetables, nuts, healthy oils, whole grains, and legumes [29,30].

## 3. Potassium and Bone

Several studies have found an inverse relationship between K intake and blood pressure [31], with a number of investigations showing the favorable effects of a K-rich diet on cardiovascular disease, in part independently of its effect on blood pressure [12,31]. Moreover, recent data indicate a favorable effect of K intake on additional outcomes [31,32]: the results of a meta-analysis suggest that K supplementation is associated with improved endothelial function—the higher the K intake, the greater the vasodilation [33]; likewise, another meta-analysis shows that the habitual dietary K intake is associated with the risk of type 2 diabetes in the general population, exhibiting a J-shape relationship [34].

On the other hand, the role of K-salts in mineral metabolism has been studied for a long time, in particular for the effects that K-bicarbonate and K-citrate may have in tubular reabsorption, but the results of all these trials appear to be conflicting.

Since 1991, Lemann et al. studied the effect of the administration of K-salts on the renal tubular phosphate transport in patients already affected by hypercalciuria [35]. The study found that the administration of both K-citrate and K-bicarbonate causes an increase in serum phosphorus concentration and a decrease in 1,25(OH)_2_ vitamin D compared to controls during the time of supplementation [35]. In another study, the same authors demonstrated that the administration of K is accompanied by a relative and an absolute decrease in urinary calcium excretion, while dietary K deprivation increases urinary calcium excretion [36]. Indeed, eight patients underwent a 5-day K deprivation diet, resulting in a significant rise of 24 h urinary calcium excretion. Subsequently, returned to their normal K intake, obtaining a normalization in urinary calcium excretion [36].

Three trials have evaluated the effect of K-salts supplementation on bone health. In 2006, Jehle et al. studied the impact of K-salts supplementation on both biochemical markers of bone turnover and bone mineral density (BMD), measured by Dual X-ray energy absorptiometry (DXA), in a group of post-menopausal women [37]. In particular, they compared the effect of 30 mmol per day dose supplementation of both K-citrate and K-chloride on bone parameters in 161 post-menopausal women with osteopenia for 1 year of treatment. They found out that the patients administered K-citrate showed a significant reduction in calcium excretion, as well as an increase in urinary citrate excretion. Notably, these findings are believed to improve bone health through an increase in serum pH, which will be discussed in the following pharagraf. In the same group, the authors observed a reduction in the marker of bone reabsorption, namely urinary hydroxyproline, and a rise in the marker of bone formation, named osteocalcin. Moreover, spine, femoral neck, and hip BMD showed a higher increase in women treated with K-citrate than in those treated with K-chloride, demonstrating that the former may be more effective than the latter with regard to bone mass. However, a strong limitation of this study was the lack of a placebo group, which did not allow to determine whether K-citrate would have provided any benefit over no treatment at all [37].

Another 2-year study on 276 post-menopausal women compared the effects on BMD of both supplementation of K-citrate, at a dose of 55.5 mEq per day, and a diet containing 300 g of additional fruits and vegetables, with a placebo group. The diet intervention was designed to have the same load as the lower K citrate group, but the kinds of fruits and vegetables were not controlled or pre-established. Given these limitations, the study did not provide any benefit of dietary intervention on BMD [38].

In the last trial in 2013, 201 elderly men and women were treated for 2 years with 60 mmol per day of K-citrate, along with calcium (500 mg per day) and vitamin D3 (1000 UI per day), and compared to patients on a placebo. In the group treated with K-citrate, BMD significantly increased by 1.7 ± 1.5% over the placebo at the spine level and by 1.6% at the femoral neck. K citrate also had positive effects on volumetric BMD (measured by Computed tomography scanning evaluation) for both dominant and non-dominant radius and tibia (BMD 1.3 ± 1.3% at tibial trabecular level) [39].

## 4. Western Diet and Bone

The transition from a pre-agriculture diet, characterized by the high consumption of vegetables and fruit and, consequently, a high K intake, to today’s consumption led to reduced K intake in the current WD pattern. The WD pattern is characterized by a high daily intake of saturated fats, sugars, salts, proteins from red meat, and refined carbohydrates [40]. With the increase in consumption of processed or pre-packaged foods, high-sugar drinks, candy, and sweets came the concomitant reductions (often dramatic) in fruits and vegetables, hence in K intake. These reductions are, in many populations, also mirrored by high sodium intake. Moreover, these diets are characterized by a high consumption of acid-generating food, typical of the WD, which promotes a chronic state of low-grade metabolic acidosis, due to the excessive number of grains in relation to the content of fruit and vegetables [15].

Diet alone can be associated with changes in BMD. Indeed, the WD is low in vitamin D, calcium, and folic acid, suggesting it may be a possible cause of a low BMD (Figure 2).
The WD is characterized by a high acid load and low K intake. The acid excess acts on bone health in two different ways. The first occurs rapidly through an ion exchange, in which bone cells are not involved (see below).Also, WD is characterized by high sodium intake, which promotes sodium and calcium kidney loss by a passive mechanism. This is responsible for osteoclastic activation and in the end, higher bone loss.The WD is also associated with an increased risk of metabolic syndrome (MetS) and diabetes, which are both characterized by increased insulin resistance and increased insulin secretion. MetS is a risk factor for fractures and low BMD in women examined by DXA for suspected osteoporosis (Op) [41]. Regarding diabetes, there is much evidence about a higher risk of fractures in patients affected by diabetes mellitus type 2, even though their BMD measured by DXA is apparently higher than that of normal controls [41]. Speculations can be made to explain insulin excess. Insulin has been proven to stimulate DNA synthesis, to promote the secretion of osteocalcin, collagen, and insulin grown factor 1 (IGF 1), and to promote the expression of RUNX2 that is involved in the osteoblastic differentiation [42]. All of these processes may decrease with the WD, affecting the BMD. However, the same mechanisms fade in non-diabetic subjects, leaving a gap in the knowledge and forcing one to find other pathways by which the WD may play a role on bone health [43,44,45]. In this regard, the WD is directly associated with increased secretion of markers of inflammation, such as *C*-reactive protein, Interleukin-6, E-selection, soluble intercellular adhesion molecule 1, and soluble vascular cell adhesion molecule 1, which promotes post-menopausal bone loss [46]. Furthermore, patients with type 2 diabetes mellitus revealed a direct relation between serum insulin grown factor 1 (IGF 1) levels and BMD. Since diabetes is related to lower IGF 1, the study concluded that there was a higher risk of fragility fractures [47].

Lastly, the effect of WD on bone health was confirmed by two experimental studies that showed that rats fed with a WD had a lower BMD. In one study, rats fed with a WD had markedly lower femur BMD and lower biomechanical properties than rats fed with a control diet rich in calcium (calcium: potassium ratio 1.0/0.8), cholecalciferol, folic acid, fiber, and carbohydrates [48]. In another study, the WD has been proven to enhance osteoclastic function and reduce osteoblastic differentiation, ultimately resulting in a low BMD [49].

## 5. Low-Grade Metabolic Acidosis

Acid–base homeostasis in the body is tightly controlled to maintain pH levels ranging between 7.35 and 7.45. This is obtained by buffering and neutralizing systems, operated by plasma proteins, bicarbonate, and other tissue proteins, such as bone itself. First of all, kidneys are implicated in the reabsorption of bicarbonate and the excretions of protons. Then, the lungs excrete carbon dioxide by breathing, reducing the levels of cations. Also, the hydrogen/potassium protonic pump located on the red cellular surface maintains the pH levels and hydro-electrolyte balance by exchanging one hydrogen for one K, increasing the K concentration in case of acidosis and lowering it in case of alkalosis. When the acid load exceeds the capacity of these systems, a higher level of hydrogen and lower levels of bicarbonate are obtained, even if the range levels of both are considered to be normal. This is known as low-grade metabolic acidosis. Some diets, such as the WD, can be a typical cause of low-grade metabolic acidosis. Aging can also be associated with renal function decline, which worsens this systemic state of low-grade metabolic acidosis in turn [50].

The acid excess acts on bone health by two different mechanisms (Figure 3).

The first mechanism occurs rapidly through an ion exchange on the bone tissue’s surface, in which bone cells are not involved. Subsequently, the protons from the systemic acidosis stimulate osteoclasts, causing reabsorption of bone and so a loss in BMD. This is a tardive mechanism, which requires cellular activation. The most promoted hypothesis for a lower BMD loss because of dietary K is through its effect on acid–base balance, although the role of the skeleton in regulating pH is debated. Indeed, alkaline calcium salts located in bones buffer the acidic pH, leading in this way to bone loss. Subsequently, K-salts protect the bones from further BMD loss. This improvement can be obtained by diets rich in fruits and vegetables and by K supplements, such as K-citrate, but not K-chloride (as seen above).

Regarding the cellular mediated mechanism, in vitro and in vivo studies state that in severe metabolic acidosis, the rise in extracellular acid concentration promotes an increase in osteoclastic activity and a decrease in osteoblast activity [53,54].

The acid load produced by the WD corresponds to 50–70 mEq every day [55]. The bone-wasting effect of the low-grade metabolic acidosis has been confirmed by a controlled study in post-menopausal women, in which high intake of some specific fruits and vegetables, such as prunes, citrus fruits, onions, broccoli, and Chinese cabbage, was compared to the consumption of other more common fruits and vegetables. The specific fruits and vegetables group presented with a greater effect on bone metabolism, in terms of BMD, compared to the control. This was mainly explained by the different acid load produced by each diet. The first group presented with a potential acid load of −23 mEq/day, i.e., alkalizing, while the control group presented with −3 mEq/day, i.e., neutral [56].

Furthermore, recent investigations show that the WD in young adult subjects and the subsequent established mild chronic metabolic acidosis increases bone reabsorption by rising serum cortisol levels. The mild hyper-glucocorticoidism is then also responsible for an osteoporotic state [57]. In this regard, glucocorticoids (GCs) impact on the entire bone homeostasis by reducing the osteoblasts and stimulating osteoclasts, both in number and function. Indeed, GCs cause increased expression of colony-stimulating factor and of receptor activators of nuclear factor kappa B ligand, and a decreased expression of osteoprotegerin, its decoy receptor. In this way, GCs stimulate the osteoclastogenesis and osteoclasts activity, promoting bone reabsorption and, as a result, a lower BMD. On the other side, GCs reduce the number of osteoblasts by repressing the osteogenic commitment of stromal progenitor cells, then diverging them into adipogenesis [58]. Latterly, the induced osteocytes apoptosis has also been proven to be induced by GCs over production [59]. Regarding the activity, GCs decrease the anabolic function of osteoblasts, the secretion of osteoid matrix proteins (collagen and osteocalcin) and subsequent mineralization of the matrix itself [58].

## 6. Discussion and Conclusions

The blood pH in humans ranges from 7.35 to 7.45 and it is tightly controlled by the mechanisms involved in the acid–base homeostasis. A chronic state of mild metabolic acidosis, even in healthy people, is frequently associated with the so-called WD [60]. The high intake of animal proteins increases the acid load, whose buffering may exceed the neutralizing capacity of the kidneys (protons excretion) and the lungs (carbon dioxide elimination). Therefore, a long-standing acidotic state brings into play the bone that contributes to the buffering process through an increase in osteoclastic resorption, which in turn removes basic substances contained inside the bone mineral phase. Together with the bases, calcium is resorbed from inside the bone and it is consequently excreted in the urine, possibly inducing hypercalciuria.

Clearly, a diet rich in fruits and vegetables may provide an alkaline environment that contributes to the buffering of protons. Nevertheless, due to the high acid load of the WD, the basic substances supplied by correct nutrition cannot prevent the increase in bone resorption.

Several studies have investigated the effects of alkaline salts on bone metabolism. Both K-citrate and K-bicarbonate have shown positive effects in this setting that are exerted through a reduction of calcium excretion, of net acid excretion, and of bone turnover markers [61]. Some papers have also found a positive effect of alkaline salts on bone mineral density, even though these findings have not always been confirmed [62].

Combined together, the results of these studies provide an encouraging picture of the benefits of alkalinization, although its real effect on bone and mineral metabolism is honestly small.

The use of K supplementation with slow-release K salts is generally safe at a low dose [63]. Hyperkalemia following excessive dietary K intake is rare in healthy individuals, and it is more likely to occur in individuals with compromised renal functions [64] or very high intakes of oral K supplements.

The high heterogeneity of the studies (enrolment of different populations, use of multiple schedules for alkaline salts administration and various durations, and the lack of pre-established dietary protocol) is probably the cause of this small effect. Notwithstanding, a high consumption of fruits and vegetables, together with the use of alkaline salts in selected cases, may provide an extra number of basic substances, such as K (bicarbonate and citrate), that represent a promising prevention of bone metabolism alterations induced by the WD.

## Figures and Tables

**Figure 1 nutrients-16-03016-f001:**
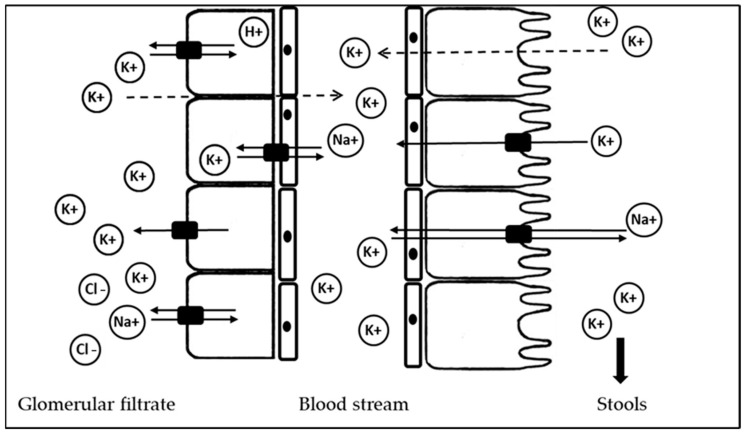
Mechanism of reabsorption of potassium by kidney and gut. Dotted line: passive mechanism of reabsorption. Continue line: active mechanism of reabsorption. K+: potassium ion. H+: hydrogen ion. Na+: sodium ion. Cl−: chlorine cation.

**Figure 2 nutrients-16-03016-f002:**
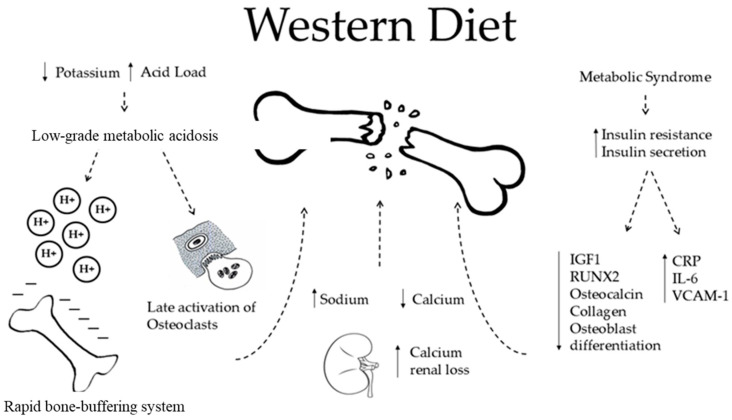
The impact of Western Diet on bone health. H^+^: Hydrogen ion; IGF1: Insulin-like growth factor 1, also known as somatomedin C; RUNX2: Runt-related transcription factor 2; CRP: C reactive protein; IL-6: interleukin-6; VCAM-1: vascular cell adhesion molecule 1; ↓: reducing; ↑: increasing; ⇣: cause-effect relationship. A Western diet acts on bone loss through some different mechanisms, causing nephrolithiasis and reduction in bone mineral density, osteoporosis and, in the end, high risk of fragility fractures [40].

**Figure 3 nutrients-16-03016-f003:**
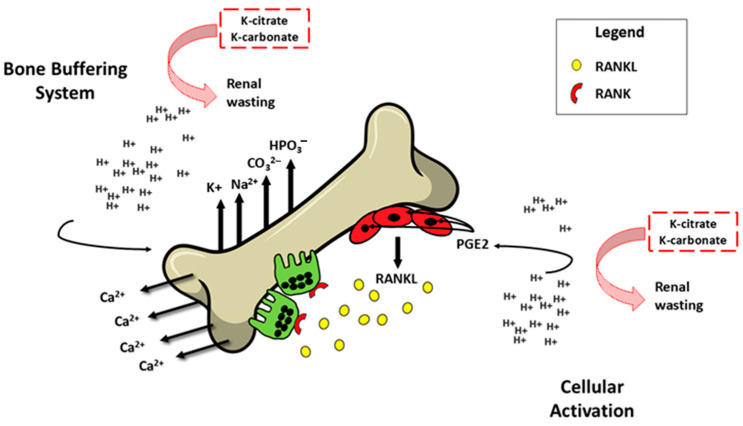
The effect of low-grade metabolic acidosis on bone health. H^+^: Hydrogen ions; Ca^2+^: calcium bication; K+: potassium cation; Na^2+^: sodium bication; CO_3_^2−^: carbon oxoanion; HPO_3_^−^: phosphoric acid; PGE2: Prostaglandin E_2_; RANKL: receptor activator of nuclear factor kappa-Β ligand; K-citrate and K-carbonate: salts of potassium. The figure represent the effect of low-grade metabolic acidosis on bone. The H+ excess is buffered by an ion-exchange on its surface on one side, representing the bone buffering system, and by cellular activation on the other. Regarding the latter, osteoclast final activation is induced by RANK (red half circle-shaped)—RANKL (red oval-shaped figure) signaling, triggered by PGE2 in turn [51]. The eventual presence of K-salts can neutralize H+ and improve calcium balance [52].

**Table 1 nutrients-16-03016-t001:** Examples of foods with a higher potassium content per 100 g [16].

Foods	K Content (mg/100 g)
Grains and cereal products	
Wheat bran	1160
Buckwheat	450
Oatmeal	370
Corn	287
Brown rice	214
Pearl barley	120
Legumes (dried * vs. cooked)	
Soy	1740 vs. 590
Bean	1445 vs. 1273
Peas	990 vs. 444
Lentils	980 vs. 185
Chickpea	881 vs. 581
Fava beans	236 vs. 228
Vegetables	
Parsley	670
Potatoes	600
Garlic	600
Spinach	530
Arugula	468
Brussels sprouts	450
Fruits	
Peaches	950
Grape	864
Apples	730
Avocado	450
Kiwi	400
Currant	370
Banana	350
Nut	
Pistachios	972
Peanuts	680
Pecan nuts	603
Cashew nuts	565
Hazelnut	466
Macadamia	363
Milk and dairy products	
Cow milk	150
Goat yogurt	251
Dairy yogurt	185
Sheep milk	182
Goat milk	180
Cream cheese	150
Drinks and Beverages ^1^	
Coffee [17]	89 to 154
Orange juice [18]	200
Soft high-sugar drinks [19]	4.3
Beer [20]	50
Wine [21]	99
Fish and derivates	
Stockfish (dried) *	1500
Anchovies in oil	700
Trout	530
Mackerel fish	360
Cod	330
Salmon	310
Meat and derivates	
Bresaola	505
Chicken	497
Turkey	475
Ham	454
Pork	370
Veal	360

K: potassium. * Higher content of potassium derives from the dried nature of the food. ^1^ mg/100 mL.

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
