# Peer review of "Potassium Intake and Bone Health: A Narrative Review"

_nutrients, 2024, doi:10.3390/nu16173016_

Round 1
Reviewer 1 Report
Comments and Suggestions for Authors
The authors have done a solid job summarizing the evidence on the relationship between potassium and bone health. The topic is relevant, given the growing interest in dietary factors that influence bone health. However, a few areas could be improved to enhance clarity and accuracy.
-
There are a few typographical errors that should be corrected (e.g. in Figure 2, “Insuline” should be corrected to “insulin").
-
Figure 2 contains several acronyms that are not explained. Please provide a legend or footnotes to explain the acronyms presented in the figure.
-
Please ensure the accuracy of the references cited in the paper. For example, reference 40 appears to be incorrectly cited and should likely be reference 41. A thorough review and verification of all references is needed.
-
Some specific statements require proper citation. For instance, in line 215, the statement, “Insulin has been proven to stimulate DNA synthesis, to promote the secretion of osteocalcin and collagen, insulin-grown factor 1 (IGF-1), RUNX2, that is involved in osteoblastic differentiation,” should be supported by a reference.
-
The discussion on the Western diet (WD) and bone health is interesting, but the relationship between the WD and potassium is not clearly articulated. The WD affects bone health through multiple pathways, and potassium's specific role within this context is not emphasized. I suggest shortening this section, focusing on the potassium content in the WD, and merging it with the subsequent section on metabolic acidosis. This will create a more coherent narrative and strengthen the connection between dietary potassium, metabolic acidosis, and bone health.
-
In the same section, it is important to acknowledge that there is no consistent clinical evidence on the relationship between insulin resistance and bone health (see DOI: 10.1210/jc.2018-02539, DOI: 10.1186/s12891-023-06817-9, DOI: 10.1002/jbm4.10780). This should be reflected in the discussion.
-
Finally, rather than focusing on the relationship between the WD and bone health, I recommend creating a figure that illustrates the relationship between low-grade metabolic acidosis and bone health, with a greater emphasis on the role of potassium. This figure would align more closely with the primary focus of the review and provide a visual summary of the key mechanisms discussed in the paper.
Overall, this review offers valuable insights into the relationship between potassium and bone health. I believe that addressing the above-mentioned suggestions will enhance the paper's clarity and impact.
Comments on the Quality of English LanguageThe overall quality of the English language is acceptable. There are a few typographical errors that should be corrected.
Author Response
The authors have done a solid job summarizing the evidence on the relationship between potassium and bone health. The topic is relevant, given the growing interest in dietary factors that influence bone health. However, a few areas could be improved to enhance clarity and accuracy.
We thank the reviewer 1 for the kind comment on our work and we appreciate the time spent on it.
- There are a few typographical errors that should be corrected (e.g. in Figure 2, “Insuline” should be corrected to “insulin").
We thank you for the notes. We corrected the mistakes.
- Figure 2 contains several acronyms that are not explained. Please provide a legend or footnotes to explain the acronyms presented in the figure.
We wrote at length all the acronyms from the Figure 2.
- Please ensure the accuracy of the references cited in the paper. For example, reference 40 appears to be incorrectly cited and should likely be reference 41. A thorough review and verification of all references is needed.
Thank you for your comment. We revised carefully the entire reference list, with appropriate links.
- Some specific statements require proper citation. For instance, in line 215, the statement, “Insulin has been proven to stimulate DNA synthesis, to promote the secretion of osteocalcin and collagen, insulin-grown factor 1 (IGF-1), RUNX2, that is involved in osteoblastic differentiation,” should be supported by a reference.
A reference for the statement has been added in the reference list.
- The discussion on the Western diet (WD) and bone health is interesting, but the relationship between the WD and potassium is not clearly articulated. The WD affects bone health through multiple pathways, and potassium's specific role within this context is not emphasized. I suggest shortening this section, focusing on the potassium content in the WD, and merging it with the subsequent section on metabolic acidosis. This will create a more coherent narrative and strengthen the connection between dietary potassium, metabolic acidosis, and bone health.
According to the suggestion of the reviewer, we shortened the Western Diet section to be more consistent with the paper.
- In the same section, it is important to acknowledge that there is no consistent clinical evidence on the relationship between insulin resistance and bone health (see DOI: 10.1210/jc.2018-02539, DOI: 10.1186/s12891-023-06817-9, DOI: 10.1002/jbm4.10780). This should be reflected in the discussion.
We thank the reviewer for the observation. Indeed, it was introduced into the Western Diet section, accompanying the discussion about diabetes and insulin and distinguishing the effect that insulin may play in diabetic and in non-diabetic subjects.
- Finally, rather than focusing on the relationship between the WD and bone health, I recommend creating a figure that illustrates the relationship between low-grade metabolic acidosis and bone health, with a greater emphasis on the role of potassium. This figure would align more closely with the primary focus of the review and provide a visual summary of the key mechanisms discussed in the paper. Overall, this review offers valuable insights into the relationship between potassium and bone health. I believe that addressing the above-mentioned suggestions will enhance the paper's clarity and impact.
We thank the reviewer for the suggestion. Another figure (figure 3) has been added to the paper. It contains a more central role of potassium and it focuses on the mechanism expressed in the paper about low-grade metabolic acidosis and bone health.
Reviewer 2 Report
Comments and Suggestions for Authors
The authors described the dynamics of potassium in the human body and its importance in bone metabolism, as well as recalling the transition of dietary potassium intake in the human species. While the importance of calcium and iron among the essential minerals is easily imagined and well known by the public, it is less well known that potassium is a mineral that contributes to good health. It should be noted that potassium is widely found in a variety of foods and has a range of favourable effects. Understanding the function of potassium in bone health may also be helpful when directing the quality of the diet. I think this review is well structured as a whole, but would like to make a few comments.
The amount of potassium in foods is given in Table 1, but the potassium content as given in the table does not appear to be described in the reference [16]. Please check.
Some standard food table should be referenced.
The authors describe Western diets as diets that are likely to be deficient in potassium, but Table 1 does not seem to list many typical Western foods.
Grains and cereal products: Typecal starch foods in WD (white bread, pasta,, ) should also be listed as examples of foods that are likely to cause potassium deficiency. Is soy flour necessary as an example in this group?
Legumes; The potassium contents in this group seem to be dried peas. The values for cooked (watered) peas may be useful.
Vegetables: Is “black pepper” suitable? The potassium content (1260mg/100g) is of course high, however, normal habitual use may be less than 1g.
Milk and dairy products: The potassium content for cow milk (1650mg/100g) seems too high. The value is 150mg/100g in the USDA food database. Please check.
Fish and derivates: Stockfish is dried, that’s why the high content of potassium. At least an annotation is required.
L89-90 “K can be found in particular in starchy roots or tubers, in vegetables, fruits, whole grains, dairy products, and even in coffee.” However, no mention of coffee or other drinks was made in Table 1.
Reference to actual sources of potassium intake (food contribution) obtained from dietary survey could also be provided.
Author Response
The authors described the dynamics of potassium in the human body and its importance in bone metabolism, as well as recalling the transition of dietary potassium intake in the human species. While the importance of calcium and iron among the essential minerals is easily imagined and well known by the public, it is less well known that potassium is a mineral that contributes to good health. It should be noted that potassium is widely found in a variety of foods and has a range of favourable effects. Understanding the function of potassium in bone health may also be helpful when directing the quality of the diet. I think this review is well structured as a whole, but would like to make a few comments.
We thank the reviewer for the kind compliments and we’ll follow the instruction as below.
The amount of potassium in foods is given in Table 1, but the potassium content as given in the table does not appear to be described in the reference [16]. Please check.
We thank the reviewer for the closer observation. The reference was wrongly added to the title of Table 1.
Some standard food table should be referenced.
References have been added to the table.
The authors describe Western diets as diets that are likely to be deficient in potassium, but Table 1 does not seem to list many typical Western foods.
Many other food typical from Western Diet has been added in the table 1 according to the suggestion.
Grains and cereal products: Typecal starch foods in WD (white bread, pasta,, ) should also be listed as examples of foods that are likely to cause potassium deficiency. Is soy flour necessary as an example in this group?
We agree with the observation. We deleted soy flour as not typical from WD.
Legumes; The potassium contents in this group seem to be dried peas. The values for cooked (watered) peas may be useful.
The amount of potassium in each cooked or watered legume has been added to the table, next to the dried content
Vegetables: Is “black pepper” suitable? The potassium content (1260mg/100g) is of course high, however, normal habitual use may be less than 1g.
According to the suggestion, black pepper was deleted to avoid any confusion while reading the Table 1.
Milk and dairy products: The potassium content for cow milk (1650mg/100g) seems too high. The value is 150mg/100g in the USDA food database. Please check.
The mistake has been corrected.
Fish and derivates: Stockfish is dried, that’s why the high content of potassium. At least an annotation is required.
We thank the reviewer for the observation. The notation has been added in the Table itself.
L89-90 “K can be found in particular in starchy roots or tubers, in vegetables, fruits, whole grains, dairy products, and even in coffee.” However, no mention of coffee or other drinks was made in Table 1.
Another line about coffee and drink has been added in table 1 according to the reviewer suggestion
Reference to actual sources of potassium intake (food contribution) obtained from dietary survey could also be provided.
References have been added to the table.
Round 2
Reviewer 1 Report
Comments and Suggestions for Authors
The authors have addressed my concerns
Reviewer 2 Report
Comments and Suggestions for Authors
The manuscript was revised appropriately.